# Spatial Patterns in Fish Assemblages across the National Ecological Observation Network (NEON): The First Six Years

Dylan Monahan [1,2,*], Jeff S. Wesner [3], Stephanie M. Parker [2] and Hannah Schartel [2]

1   River Rat Watershed Solutions, Boulder, CO 80026, USA
2   National Ecological Observatory Network, Battelle, Boulder, CO 80301, USA
3   Biology, University of South Dakota, Vermillion, SD 57069, USA; jeff.wesner@usd.edu
*   Correspondence: river.rat.h20shed@gmail.com

**Abstract:** The National Ecological Observation Network (NEON) is a thirty-year, open-source, continental-scale ecological observation platform. The objective of the NEON project is to provide data to facilitate the understanding and forecasting of the ecological impacts of anthropogenic change at a continental scale. Fish are sentinel taxa in freshwater systems, and the NEON program has been sampling and collecting fish assemblage data at wadable stream sites for six years. One to two NEON wadable stream sites are located in sixteen domains from Alaska to Puerto Rico. The goal of site selection was that sites represent local conditions but with the intention that site data be analyzed at a continental observatory level. Site selection did not include fish assemblage criteria. Without using fish assemblage criteria, anomalies in fish assemblages at the site level may skew the expected spatial patterns of North American stream fish assemblages, thereby hindering change detection in subsequent years. However, if NEON stream sites are representative of the current spatial distributions of North American stream fish assemblages, we could expect to find the most diverse sites in Atlantic drainages and the most depauperate sites in Pacific drainages. Therefore, we calculated the alpha and regional (beta) diversities of wadable stream sites to highlight spatial patterns. As expected, NEON sites followed predictable spatial diversity patterns, which could facilitate future change detection and attribution to changes in environmental drivers, if any.

**Keywords:** NEON; fish; diversity; assemblages

**Key Contribution:** NEON sites have collected fish assemblage data for six years. Currently, NEON sites follow a predictable spatial pattern of fish assemblage diversity.

## 1. Introduction

Fish assemblage data can quantify the variety of fish species across any area and can provide important information about land use, water pollution, habitat degradation, and invasive species. Using fish assemblages to assess human-caused stream degradation requires an understanding of the expected fish assemblage for that site or region [1–4].

The National Ecological Observation Network (NEON) is a thirty-year, open-source, continental-scale ecological observation platform. The objective of the NEON project is to provide researchers and the public with data to facilitate an understanding and forecasting of the ecological impacts of climate, land use changes, and invasive species at a continental scale. NEON open-source data allow researchers to access continental data collected using uniform protocols. NEON provides infrastructure and consistent methodologies for the collection and analysis of these data [5]. Consistent observations at a continental extent can help users compare smaller-extent watershed studies to broader extents [6].

The NEON freshwater program is a vital part of NEON's goal of detecting and quantifying the drivers of ecological change by sampling community composition, measuring surface and groundwater chemistry, deploying micrometeorology and in situ water quality instrumentation in and around water bodies, and tracking habitat structures [5].

Fish assemblages are an important component of freshwater ecosystem data. Fish are considered sentinel taxa in freshwater systems because they are often mobile and play essential roles in energy and nutrient transfer. Therefore, quantitative fish data are an important component in detecting aquatic ecosystem patterns and changes (4). NEON fish sampling methods provide fish assemblage data, which are a vital tool for researchers now and in the future.

A foundational principle of the NEON program is that fish assemblages are a useful indicator of anthropogenic influences. NEON stream sites are collocated with other environmental data, allowing users to better understand the drivers of changes to fish assemblage data [7–9]. A challenge to using NEON fish assemblage data is that fish assemblages need to be understood at both the site and biogeographic levels to assess the effects of potential anthropogenic degradation. NEON's wadable stream sites were selected to answer a broad range of ecological questions at varying scales, but were not specifically selected to represent the full range of regional fish assemblages.

Nonetheless, a key question is as follows: do NEON sites represent expected continental-scale fish assemblage patterns? Fish assemblages in wadable stream sites are determined by several site-specific features: these include where the site is located, habitat conditions, and the historical pattern of fish colonization at the site [10,11]. The macroecological context within which a site sits is often an important fish assemblage predictor. In the United States, primarily due to glacial history and climate change, we would expect to see the highest alpha diversities in Atlantic drainage sites, particularly in sites found in warmer lowland river drainages [12]. Sites in Pacific drainages with colder winters, drier summers, and less stable river drainages would be expected to have the lowest biodiversity scores [13]. We would also expect beta diversity to show an effect on the drainage location when comparing site dissimilarity.

Here, we use alpha and beta diversity metrics and size composition data from NEON wadable stream sites to describe spatial patterns in NEON fish data. In describing these spatial patterns, we seek to confirm whether NEON sites in the first years of sampling (2017–2022) meet expected spatial fish assemblage patterns. We also check size composition to contextualize the ecological relationship of assemblages to their sites in a manner comparable to a continental scale. We use species occurrence data to inform potential differences in seasonal and temporal sampling.

## 2. Materials and Methods

### 2.1. NEON Site Selection

The NEON observatory is divided into 20 ecoclimatic domains based on statistical geographic clustering [14]. Fish were collected at 23 wadable stream sites in 16 domains, (Figure 1; Table 1; see neonscience.org for additional site information, accessed between 1 January 2017 and 31 December 2022). NEON does not sample fish at large river sites because the effort needed to conduct quantitative fish sampling in rivers exceeds NEON's resources.

### 2.2. Biological Sampling Windows

Fish sampling at NEON sites occurs twice per year, annually, in the spring and fall. Because of the wide seasonal range of sites spread out from Alaska to Puerto Rico, spring and fall cover a range of months depending on the site (Table 2). Spring sampling dates are intended to coincide with the start of warming degree days and the start of peak greenness, and fall sampling dates are determined to coincide with a decrease in light levels and temperature at the site. These criteria were chosen as the fish sampling windows, as they are an important biogeochemical catalyst and allow fish data to be associated with those collocated biogeochemical parameters [5,14].

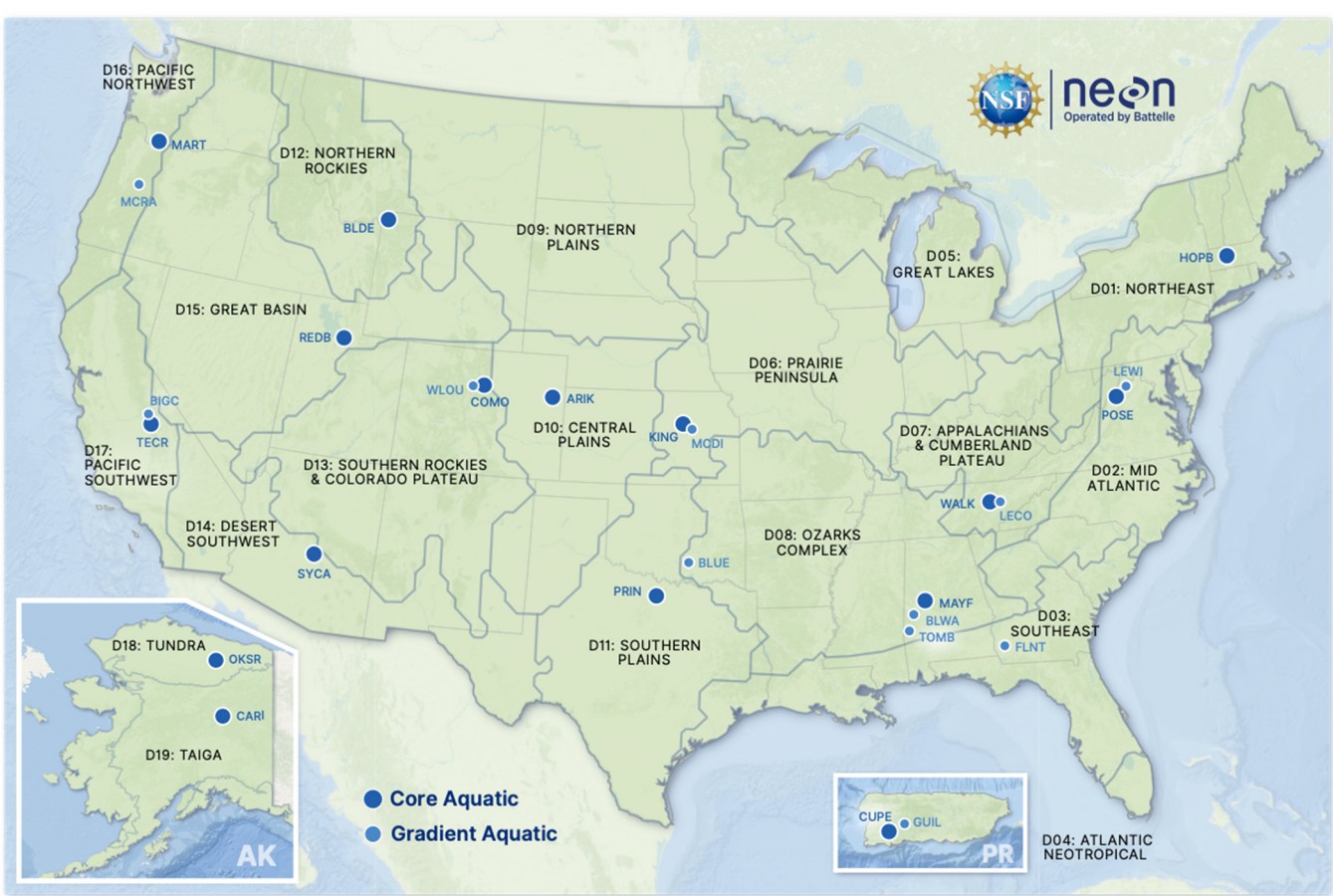

**Figure 1.** NEON wadable stream sites in 20 ecoclimatic domains. Core sites are wilderness sites, and gradient sites are sites with known anthropogenic stressors [15].

**Table 1.** NEON sites, drainages, domain numbers, and domain names.

| NEON Site Name | Drainage | Domain Number | Domain Name |
|:---:|:---:|:---:|:---:|
| HOPB | Atlantic | 01 | Northeast |
| LEWI | Atlantic | 02 | Mid-Atlantic |
| POSE | Atlantic | 02 | Mid-Atlantic |
| CUPE | Atlantic | 04 | Atlantic Neotropical |
| GUIL | Atlantic | 04 | Atlantic Neotropical |
| MCDI | Atlantic | 06 | Prairie Peninsula |
| KING | Atlantic | 06 | Prairie Peninsula |
| LECO | Atlantic | 07 | Appalachian |
| WALK | Atlantic | 07 | Appalachian |
| MAYF | Atlantic | 08 | Ozark Complex |
| MAYF | Atlantic | 08 | Ozark Complex |
| ARIK | Atlantic | 10 | Central Plains |
| BLUE | Atlantic | 11 | Southern Plains |
| PRIN | Atlantic | 11 | Southern Plains |
| BLDE | Atlantic | 12 | Northern Rockies |
| WLOU | Pacific | 13 | Southern Rockies |

**Table 1.** *Cont.*

| NEON Site Name | Drainage | Domain Number | Domain Name |
|---|---|---|---|
| SYCA | Pacific | 14 | Desert Southwest |
| REDB | Pacific | 15 | Great Basin |
| MART | Pacific | 16 | Pacific Northwest |
| MCRA | Pacific | 16 | Pacific Northwest |
| BIGC | Pacific | 17 | Pacific Southwest |
| TECR | Pacific | 17 | Pacific Southwest |
| OKSR | Pacific | 18 | Tundra |
| CARI | Pacific | 19 | Taiga |

**Table 2.** Domain site-sampling windows.

| Domain | Site | Spring Sampling Window | Fall Sampling Window |
|---|---|---|---|
| 1 | HOPB | 11 Apr–9 May | 3 Oct–31 Oct |
| 2 | POSE | 19 Mar–16 Apr | 18 Oct–15 Nov |
| 2 | LEWI | 19 Mar–16 Apr | 18 Oct–15 Nov |
| 4 | CUPE | 24 Jan–21 Feb | 10 Nov–8 Dec |
| 4 | GUIL | 26 Jan–23 Feb | 9 Nov–7 Dec |
| 6 | KING | 23 Mar–20 Apr | 3 Oct–31 Oct |
| 6 | MCDI | 20 Mar–17 Apr | 27 Sep–25 Oct |
| 7 | LECO | 15 Mar–12 Apr | 12 Oct–9 Nov |
| 7 | WALK | 09 Mar–06 Apr | 19 Oct–16 Nov |
| 8 | MAYF | 05 Mar–02 Apr | 24 Oct–28 Nov |
| 10 | ARIK | 21 Mar–18 Apr | 20 Sep–18 Oct |
| 11 | PRIN | 17 Feb–17 Mar | 23 Oct–20 Nov |
| 11 | BLUE | 07 Mar–04 Apr | 12 Oct–9 Nov |
| 12 | BLDE | 10 Jun–08 Jul | 30 Aug–27 Sep |
| 13 | COMO | 05 Jul–02 Aug | 5 Sep–3 Oct |
| 13 | WLOU | 02 Jul–30 Jul | 3 Sep–1 Oct |
| 14 | SYCA | 12 Jan–11 Feb | 3 Jun–3 Jul |
| 15 | REDB | 29 Mar–26 Apr | 29 Sep–27 Oct |
| 16 | MCRA | 10 Apr–08 May | 23 Sep–21 Oct |
| 16 | MART | 06 Apr–04 May | 22 Sep–20 Oct |
| 17 | TECR | 06 May–17 Jun | 17 Sep–15 Oct |
| 17 | BIGC | 02 Apr–30 Apr | 28 Sep–26 Oct |
| 18 | OKSR | 21 May–18 Jun | 7 Aug–4 Sep |
| 19 | CARI | 02 May–30 May | 18 Aug–15 Sep |

Sampling windows are 28 days long [14] and based on historic, publicly available air temperatures from the National Oceanographic and Atmospheric Administration (NOAA) and riparian phenology Moderate Resolution Imaging Spectroradiometer (MODIS) data. Contingent decisions include allowing fish sampling for up to 30 days after the end of the sampling window to accommodate staffing concerns, weather delays, and high or low water, as documented for the data users [14]. As more years of consecutive NEON data

become available, data used to define the sampling windows are replaced with NEON sensor and stream discharge data, allowing the sampling to be flexible with changing site or climate conditions over the lifetime of the NEON project.

*2.3. NEON Fish Data*

NEON stream sites are 1 km long and divided into 10 (80–100 m) reaches, except for MCDI, where the sampling permit restricts the site to 500 m. Six (80–100 m) reaches are scheduled for DC backpack electrofishing at each site (except MCDI, with three reaches scheduled per bout), using the NEON wadable stream fish sampling protocol at every site [16]. No major protocol changes occurred over the six years of this study. Three of the six scheduled reaches were fixed reaches sampled every visit (Figure 2) by employing three-pass depletion sampling. The other three reaches included random reaches that came from a panel of seven random reaches sampled on a rotating schedule. Each random reach was randomly selected before the first year of sampling and scheduled for sampling so that each random reach was scheduled to be sampled at least once every three years, and random reaches were sampled on a single pass. At MCDI, where land ownership and permitting the restricted site length to 500 m occurred, there were five designated reaches; therefore, each year, only two fixed and one random reach were sampled per bout per year.

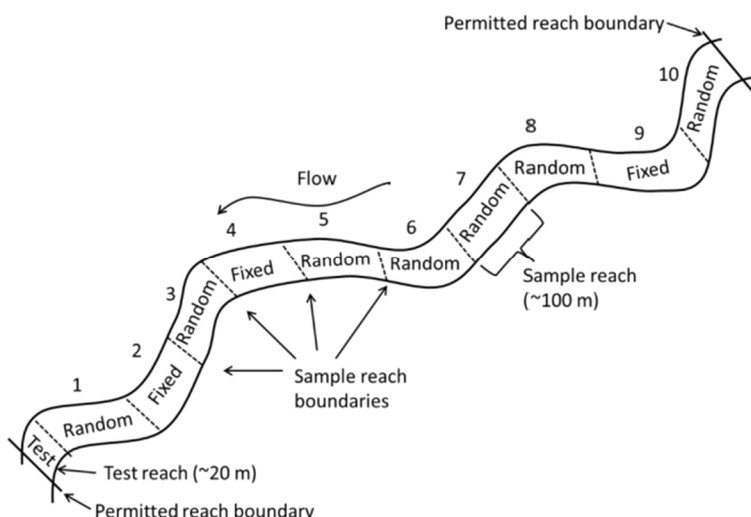

**Figure 2.** Schematic of a 1 km NEON stream site delineated into ten 100 m reaches: 3 fixed and 7 random sampling reaches. The three fixed reaches are sampled every visit; three random reaches are chosen each year for sampling [16].

All reaches were closed-sampled with fixed block nets set at the top and bottom of the reach. Not all the reaches scheduled are always sampled each bout because of weather, equipment, and logistic issues. One fixed reach per bout was the minimum effort required for fish data to be available to the public.

Captured fish were identified to the lowest taxonomic level possible based on [17] and the Integrated Taxonomic Information System (ITIS) online database (http://www.itis.gov accessed on 31 July 2017). When field scientists responsible for identification were uncertain, they used a morphospecies or identification qualifier. Some fish were identified only to their family or genus (followed by an SP. or SPP.). Federally listed species are obscured when published so that they appear identified at the family level; this protects listed species and is part of the NEON agreement with the U.S. Fish and Wildlife Service.

The first fifty individuals captured from the same taxonomic identification group per reach were wet-weighed (g) and measured to the total length (mm). After fifty individuals from the same taxonomic identification group were measured and weighed, all fish captured in that reach from that taxonomic group were bulk-counted and not measured. This process started again at the start of each reach sample.

*2.4. Downloading and Compiling NEON Fish Data*

NEON electrofishing data were downloaded on 16 April 2023 from the NEON data portal [18]. Taxonomic data were counted per bout from measured fish and bulk fish data. Only first-pass data were used unless a new species was collected in a 2nd or 3rd pass from a 3-pass depletion reach, and it was also caught at one of the single-pass reaches. The catch per unit effort was calculated and normalized to the hour for all taxonomies captured during a bout. Taxonomic data were counted per bout from the measured fish and bulk fish data.

*2.5. Alpha Diversity*

To describe the spatial distribution of fish taxonomy at NEON stream sites, the vegan R Package was used to calculate species richness and Shannon and Simpson metrics for each site on a per-bout basis [19]. All first-pass, electrofishing data of both fixed and random reaches from 2017 to 2022 were analyzed.

*2.6. Beta Diversity*

To test the diversity between drainages, beta diversity was mapped for all bouts from 2017 to 2022 for each of the 23 wadable stream sites, using the betadiver and betadisper command in Vegan [19]. CPUE data were used to calculate the dissimilarities between species observed from data via Atlantic and Pacific drainage and a principal coordinates analysis (PCoA) distribution using a beta z distribution.

*2.7. Size Composition*

NEON measures individual fish sizes (total lengths and field measured wet weight) for the first fifty individuals of each species on each electrofishing pass. The total length is measured to the nearest 0.1 mm, and wet weight is measured to the nearest 0.1 g, with a lower limit of 0.3 mg. The resulting dataset contained 52,882 individual measures of fish length and wet weight from 2015 through 2022. Lengths and wet weights were compiled and sorted by species, site, and years.

## 3. Results

*3.1. Alpha Diversity Metrics*

Since 2017, NEON has collected 112 species of fish at wadable stream sites. The greatest number of fish species sampled were from domains 01, 02, 04, 06, 07, 08, and 10, including 42 species sampled at NEON's MAYF site in domain 08 and 33 at NEON's BLUE site in domain 11. All of these domains were Atlantic draining. The lowest scores were recorded at domains 12, 13, 14, 15, 16, 17, 18, 19, and 20 (all Arctic and Mountain West sites). Except for SYCA in the Desert Southwest (domain 15), species at these lower-scoring sites were dominated by Salmonidae (Table A1).

Similarly, Shannon diversity scores were highest at NEON's Southeastern, Southern, Central Plains, Atlantic, and Caribbean sites and lowest at NEON's West Coast, Arctic, and Mountain West sites (Figure 3). Since 2017, ten sites produced fish species during spring sampling but not in fall sampling, and nine sites yielded occurrences of fish species during fall sampling but not in the spring sampling (Table A2). This indicates the usefulness of seasonal sampling, as well as its hindrance to assessing annual trends.

*3.2. Beta Diversity Metrics*

PCoA mapping shows that some site visits in Pacific drainages were similar to some of those in Atlantic drainages (Figure 4). However, Atlantic drainage sites have a greater range in the fish species found at those sites, indicating a greater level of beta diversity in Atlantic drainage sites.

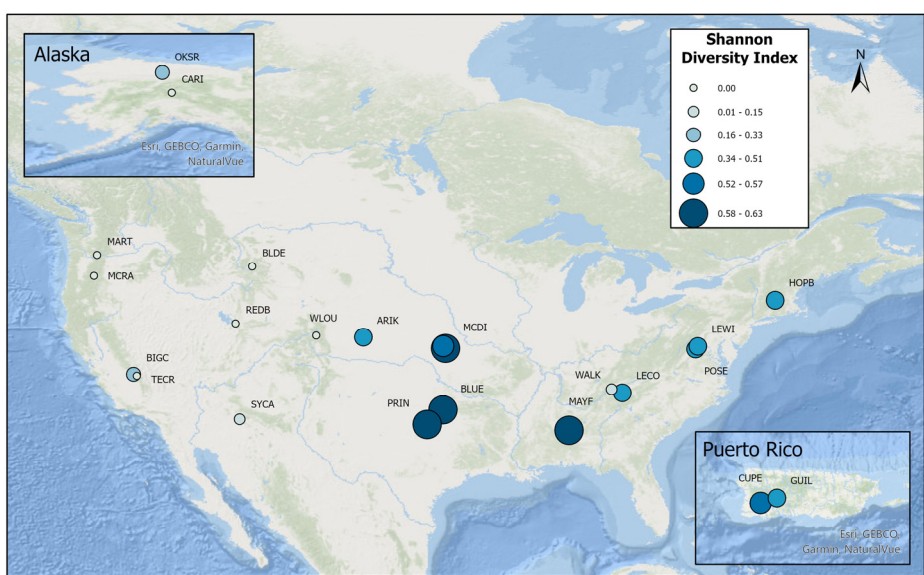

**Figure 3.** NEON stream—fish Shannon diversities.

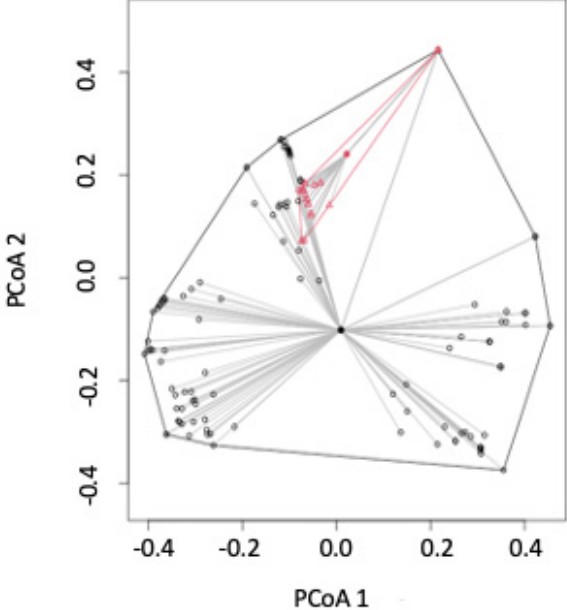

**Figure 4.** PCoA distances for fish species sampled at NEON sites from 2017 to 2022 and apportioned by Atlantic (black circle) vs. Pacific (red triangle) drainage.

*3.3. Size Composition*

Fish wet weights ranged from four orders of magnitude across all collections (Figure 5). Individual wet weights ranged from 0.3 g (*n* = >13,000 individuals from multiple species) to 1000 g (*n* = 7 *Oncorhynchus mykiss*). When averaged among sites, the median fish size varied from 0.3 to 22 g for wet weight and 31 to 144 mm for the total length (Table 3). Despite the variation in fish size among sites, there was strong consistency in fish sizes across years (Figure 6). For example, the grand median fish size at KING was 0.6 g, and yearly medians ranged only from 0.3 to 1 mg in wet weight. By comparison, the grand median at WLOU was 10 g, with yearly medians ranging from 6 to 13. In other words, fish size appeared to vary more among sites than across years within a site (Figure 6).

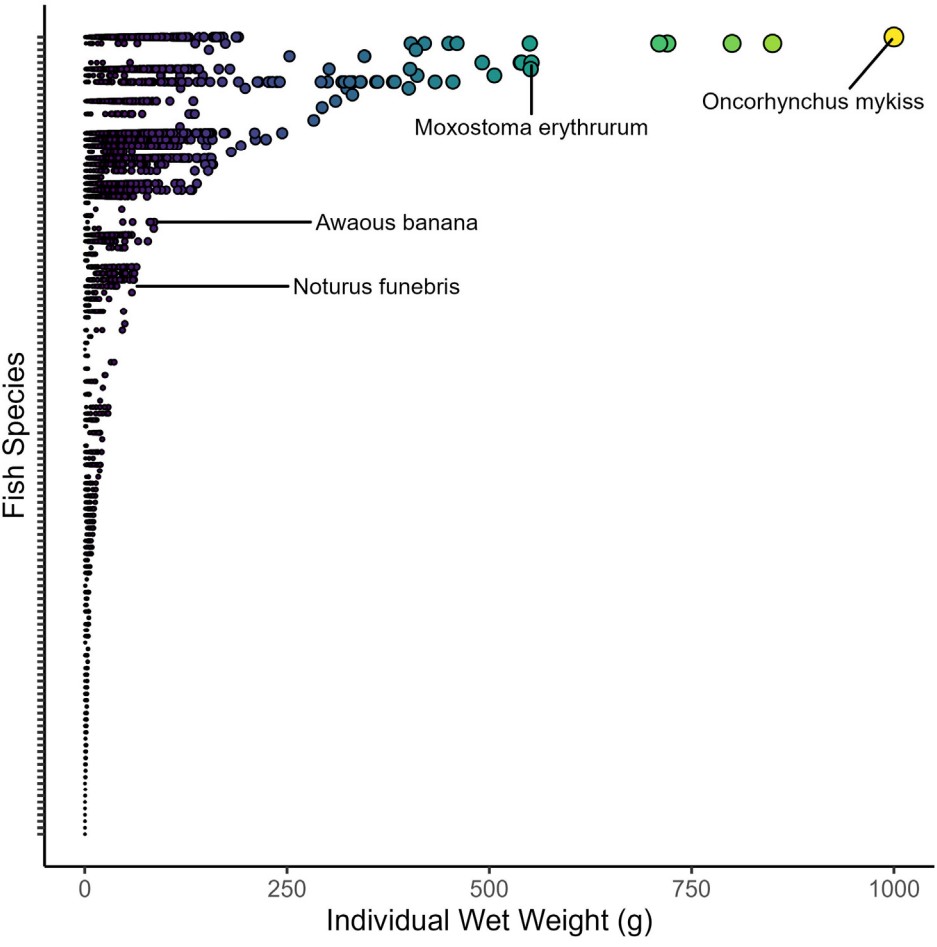

**Figure 5.** Distribution of 52,882 individual wet weights of fish measured in 23 NEON wadable stream sites. The data include all fish measured from 2015 to 2022. The y-axis represents 154 taxa ranked by the maximum fish size per taxon. Most taxon names are removed for clarity. Colors and sizes reflect the relative wet weights of fishes (yellow = largest, black = smallest).

**Table 3.** Median (and upper and lower 95%iles) of individual total lengths and wet weights summarized across all individuals collected between 2016 and 2022. N is the number of individual fish sizes recorded at a stream site.

| Site | *N* | Total Length (mm) | Wet Weight (g) |
|------|-----|-------------------|----------------|
| CARI | 186 | 144 (66 to 353) | 22.1 (3 to 361) |
| REDB | 242 | 142.5 (44 to 244) | 27.25 (1 to 140) |
| TECR | 876 | 123 (51 to 221) | 17 (0 to 95) |
| BLDE | 722 | 121 (65 to 215) | 17 (3 to 89) |
| BIGC | 2036 | 105 (31 to 215) | 11 (0 to 96) |
| WLOU | 840 | 104 (40 to 173) | 10.5 (0 to 52) |
| MCRA | 852 | 101 (41 to 158) | 8.75 (1 to 37) |
| MART | 1006 | 98 (57 to 154) | 7.9 (2 to 32) |
| LECO | 4456 | 75 (35 to 177) | 3.8 (0 to 53) |
| HOPB | 4144 | 62 (27 to 154) | 2.2 (0 to 32) |
| LEWI | 5641 | 59 (34 to 125) | 2.4 (0 to 20) |
| MCDI | 3488 | 55 (32 to 122) | 1.6 (0 to 21) |

**Table 3.** *Cont.*

| Site | N | Total Length (mm) | Wet Weight (g) |
|---|---|---|---|
| WALK | 4680 | 55 (23 to 91) | 1.4 (0 to 8) |
| MAYF | 417 | 52 (6 to 144) | 1.2 (0 to 35) |
| POSE | 6019 | 52 (24 to 89) | 1.3 (0 to 8) |
| OKSR | 180 | 50 (40 to 175) | 0.9 (0 to 39) |
| BLUE | 1519 | 45 (18 to 126) | 1.1 (0 to 25) |
| KING | 3652 | 42 (22 to 107) | 0.7 (0 to 12) |
| PRIN | 3861 | 42 (17 to 135) | 0.8 (0 to 36) |
| ARIK | 3201 | 41 (21 to 95) | 0.3 (0 to 10) |
| CUPE | 890 | 37 (14 to 220) | 0.6 (0 to 118) |
| GUIL | 1880 | 33 (13 to 82) | 0.3 (0 to 4) |
| SYCA | 2094 | 31 (17 to 67) | 0.3 (0 to 4) |

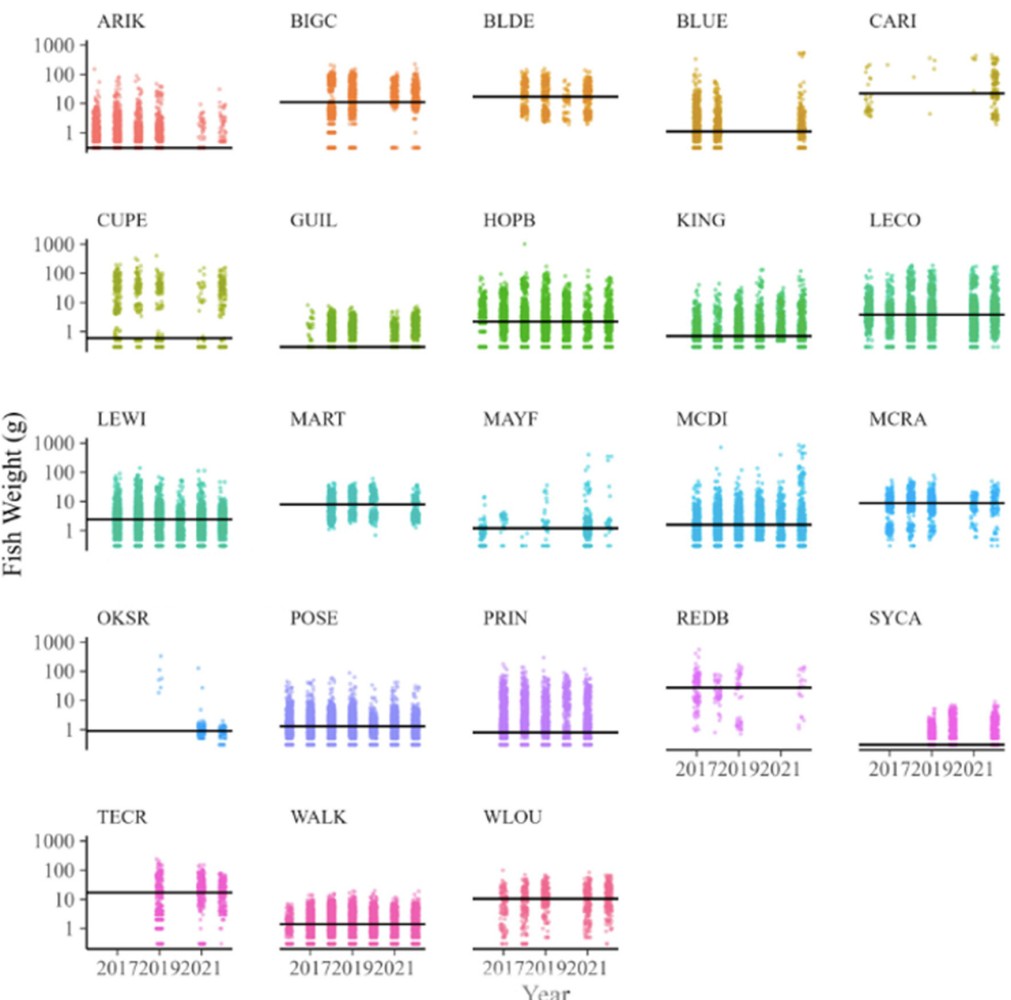

**Figure 6.** Individual fish wet weights (*n* = 52,882) collected across 23 NEON stream sites from 2017 to 2022. The horizontal line shows the grand median for each site.

## 4. Discussion

The site-level fish assemblage richness in wadable streams was determined by several site-specific features, but at North American temperate sites, one of the most important

determinants was where in the continent the site was located [10,11]. In the U.S., primarily because of glacial history and historic post-glacial fish colonization patterns, we expected to see the highest alpha and beta diversities amongst Atlantic drainage sites, particularly in those sites found in warmer lowland river drainages [12,13]. Sites in Pacific drainages where post-glacial fish colonization was much more limited were expected to have lower freshwater diversity [20]. Our results confirmed this pattern.

The NEON program is designed to monitor anthropogenic change at a continental scale. Fish assemblages are determined by spatially extensive macroecological drivers, as well as by local natural (barriers, natural disturbance) and anthropogenic (dams and species introduction) factors (4). Many studies focus on more localized drivers of fish assemblage composition, but NEON's mission is to provide data that use measures of site-specific conditions with the intention of scaling these conditions to the continental scale.

Because of the ability of local conditions to create anomalous fish assemblages in comparison to regional fish assemblages, it is important to determine whether locally collected and analyzed NEON fish assemblages represent the expected fish assemblage distribution for wadable streams at a continental scale. If NEON selection selects regionally anomalous sites where Pacific draining sites have high biodiversity and Atlantic draining sites low biodiversity, future changes in regional biodiversity caused by climate and land and water use changes may not be detected.

### 4.1. Spatial Patterns of NEON Stream Fish Data

NEON's site selection is driven by several factors, including the need to have sites represented in 19 nationwide domains. NEON sites are installed to capture local conditions but then scale up to the continental scale. Fish assemblage composition was not a factor used to select sites; instead, site selection was driven by the need to distribute sites along major continental-scale ecological gradients [15]. Because of the need to place sites on such a broad continental scale, we assumed that spatial patterns of fish assemblages would follow continental-scale diversity patterns, with the largest number of species and the highest alpha diversity metrics found at sites in the Plains, Prairies, Gulf Coast, Atlantic, and Caribbean domains, opposite of the West Coast and Mountain West domains. As expected, we found that NEON fish diversity was distributed along the expected continental-scale patterns, with the highest diversity sites found in Atlantic drainages and the lowest diversity found in Pacific drainages farther west. Also, sites west of the divide are dominated by salmonids, except for SYCA in the Desert Southwest domain, where *Agosia chrysogaster*, longfin dace, dominates. Eastern sites are dominated primarily by *Cyprinidae, Poecilidae, Cottidae*, and *Cyprinidae*.

### 4.2. Occurence

The occurrence of fish species both seasonally and by year showed that at more diverse sites, there was a higher likelihood of collecting fish in one bout but not another. This was particularly true at BLUE and MAYF. Most species caught only in one bout are relatively rare seasonally because of migratory behavior (see below). Of the 49 collection times, one species was caught at a site during only one of the seasonal bouts, but 39 of those times, it represented five fish or fewer. When comparing fish caught in either the 2017–2019 group or the 2020–2022 group, 44 out of the 68 collection times represented occurrences of five fish or less. This indicates the difficulty of collecting rare species, especially singletons and doubletons (one or two individuals) (Table A3) [21,22]. Furthermore, spring and fall sampling periods are more likely to coincide with fish migration periods, meaning natural periods of presence and absence [22–25].

### 4.3. Spatial Patterns in Size Composition

In addition to taxonomic composition and abundance, body size provides critical ecological information in relation to the age–structure, size–abundance [26] metabolism [27], food web structure [28], and trophic transfer efficiencies [29]. Arranz et al. [30] used

stream fish size spectra to detect responses to species invasion and eutrophication. Similar analyses are possible with NEON data. For example, Pomeranz et al. [31] used NEON macroinvertebrate body sizes to examine how size spectra scaled with temperature from Puerto Rico to Alaska. A major benefit of NEON size data is its collection of repeated measures over time. As shown in Figure 5, size data are consistent across the years, meaning that future disturbances to NEON sites may reveal shifts in the size structure of fish species relative to the baseline. Future data collected at NEON sites can reveal whether and how disturbances affect the size structure of fish species, yielding information not only on taxonomic persistence and abundance (from other NEON metrics) but also allowing the ecological functions of these sites to be correlated with body size [25].

## 5. Summary and Conclusions

This analysis confirms that fish assemblage patterns at NEON sites follow predicted continental-scale patterns, even though they are not selected using fish assemblage criteria. Speciose Atlantic sites were dominated by smaller-sized fish, with the most common fish representing at least five families. Low-diversity Pacific sites often contain only a single species and are dominated by salmonids. Sites representing expected spatial assemblage patterns are a good sign; potential changes to fish assemblages could be easier to detect and attribute to environmental drivers.

It is also important to learn why some fish are present in the first three years but not the last three years. Was this an accident of sampling, or are there other reasons (i.e., El Niño years vs. La Niña years)? Is it cyclical, or are those fish that were collected in the first three years gone forever? Are some the results of narrow window spawning migrations? Research in Brazil is currently examining this hypothesis at a large spatial scale (via ichthyoplankton sampling) and collaborating on an important question about how fish assemblages at a multi-continental level can maximize the benefits of NEON data [32]. The application of more sophisticated beta diversity metrics to spatial diversity questions could also be a valuable next step.

**Author Contributions:** D.M.: Introduction, Methods (Biodiversity), Results, Discussion, Conclusion. J.S.W.: Methods (Size Composition), Results (Size Composition), Discussion (Size Composition). S.M.P.: Methods (NEON Site Selection, Biological Sampling Windows). H.S.: Review documents and figures. All authors have read and agreed to the published version of the manuscript.

**Funding:** The National Ecological Observatory Network is a program sponsored by the National Science Foundation and operated under cooperative agreement by Battelle. This material is based in part upon work supported by the National Science Foundation through the NEON Program, as well as Grant No. 2106067 to JSW.

**Data Availability Statement:** Available online: https://data.neonscience.org/data-products/DP1.20107.001/RELEASE-2023 (accessed on 1 May 2023).

**Conflicts of Interest:** The authors declare no conflict of interest.

## Appendix A

**Table A1.** Species richness per site, as described by the mean number of species, spring and fall bout means, the bout mean, highest and lowest scores, most common species (the total number caught), and domain. Fish species are written in 6-letter code, with a star next to it indicating that the fish in question in not native to that site. Codes are described at the bottom of the table.

| Site | Mean | Spring Bout | Fall Bout | Highest | Lowest | Spring Bout Most Com. Spec. | Fall Bout Most Com Spec. | Domain |
|------|------|-------------|-----------|---------|--------|----------------------------|--------------------------|--------|
| BLUE | 20.75 | 17.67 | 30 | 30 | 5 | ETHRAD (786) | ETHRAD (644) | Southern Plains |

**Table A1.** *Cont.*

| Site | Mean | Spring Bout | Fall Bout | Highest | Lowest | Spring Bout Most Com. Spec. | Fall Bout Most Com Spec. | Domain |
|------|------|-------------|-----------|---------|--------|-----------------------------|--------------------------|--------|
| MAYF | 17 | 17 | 17 | 23 | 13 | NOTBAI (561) | NOTBAI (1053) | Ozarks Complex |
| PRIN | 9.5 | 10 | 9 | 11 | 7 | GAMAFF (654) | GAMAFF (1954) | Southern Plains |
| MCDI | 9.17 | 7.5 | 10.83 | 13 | 6 | CAMANO (1652) | CAMANO (566) | Prairie Peninsula |
| ARIK | 7.13 | 6.4 | 8.33 | 10 | 5 | ETHSPE (552) | * GAMAFF (1914) | Central Plains |
| KING | 6.42 | 5.33 | 7.5 | 9 | 3 | ETHSPE (790) | CHRERY (4911) | Prairie Peninsula |
| LEWI | 5.81 | 5.2 | 5.17 | 7 | 4 | COTGIR (1955) | COTGIR (2930) | Mid-Atlantic |
| CUPE | 4.4 | 4.8 | 4 | 7 | 2 | POERET (265) | POERET (111) | Atlantic NeoTropical |
| HOPB | 4.19 | 4.6 | 3 | 7 | 2 | RHIATR (674) | RHIATR (1382) | Northeast |
| POSE | 4.19 | 4.2 | 4.17 | 5 | 4 | RHIATR (1351) | RHIATR (2181) | Mid-Atlantic |
| LECO | 3.44 | 3.5 | 3.4 | 4 | 3 | RHIATR (755) | RHIATR (1965) | Appalachian and Cumberland Plateau |
| WALK | 2.46 | 2.4 | 2.5 | 4 | 2 | RHIATR (1221) | RHIATR (2811) | Appalachian and Cumberland Plateau |
| GUIL | 2.3 | 2.4 | 2.2 | 3 | 2 | POERET (3698) | POERET (4199) | Atlantic NeoTropical |
| BIGC | 2.29 | 2.75 | 1.667 | 3 | 1 | * SALTRU (645) | * SALTRU (695) | Pacific Southwest |
| SYCA | 2.25 | 2.33 | 2 | 3 | 1 | AGOCHR (2410) | AGOCHR (699) | Desert Southwest |
| OKSR | 1.33 | 2 | 1 | 2 | 1 | THYARC (1) | THYARC (64) | Tundra |
| CARI | 1.14 | 0.75 | 1.67 | 3 | 1 | THYARC (8) | THYARC (87) | Taiga |
| BLDE | 1 | NA | 1 | 1 | 1 | NA | * SALFON (587) | Northern Rockies |
| MART | 1 | NA | 1 | 1 | 1 | NA | SALSP (785) | Pacific Northwest |
| MCRA | 1 | NA | 1 | 1 | 1 | NA | ONCCLA (720) | Pacific Northwest |
| REDB | 1 | 1 | 1 | 1 | 1 | ONCCLA (24) | ONCCLA (154) | Great Basin |
| TECR | 1 | 1 | 1 | 1 | 1 | * SALFON (529) | * SALFON (387) | Pacific Southwest |
| WLOU | 1 | 1 | 1 | 1 | 1 | * SALFON (74) | * SALFON (494) | Southern Rockies and Colorado Plateau |

Etheostoma radiosum = ETHRAD, Notropis baileyi = NOTBAI, Gambusia affinis = GAMAFF, Campostoma anomalum = CAMANO, Campostoma anomalum = ETHSPE, Chrosomus erythrogaster = CHRERY, Cottus girardi = COTGIR, Poecilia reticulata = POERET, Rhinichthys atratulus = RHIATR, Salmo trutta = SALTRU, Agosia chrysogaster = AGOCHR, Thymallus arcticus = THYARC, Salvelinus fontinalis = SALFON, Oncorhynchus clarki = ONCCLA.

**Table A2.** Species caught during either the spring sampling or the fall sampling bout but not the other per-site since 2017.

| Site | Spring/Fall | Species | Count |
|------|-------------|---------|-------|
| ARIK | Spring | *Etheostoma exile* | 206 |
| BLUE | Fall | *Ameiurus melas* | 1 |
| BLUE | Spring | *Cyprinella camura* | 1 |
| BLUE | Fall | *Lythrurus umbratilis* | 6 |
| BLUE | Spring | *Micropterus punctulatus* | 1 |
| BLUE | Spring | *Micropterus salmoides* | 1 |
| BLUE | Spring | *Notropis boops* | 13 |
| BLUE | Spring | *Notropis buchanani* | 6 |
| BLUE | Fall | *Pylodictis olivaris* | 1 |
| CUPE | Spring | *Anguilla rostrata* | 3 |
| GUIL | Fall | *Tilapia rendalli* | 1 |
| HOPB | Spring | *Ameiurus nebulosus* | 1 |

**Table A2.** *Cont.*

| Site | Spring/Fall | Species | Count |
|---|---|---|---|
| HOPB | Spring | *Notemigonus crysoleucas* | 1 |
| HOPB | Spring | *Noturus gyrinus* | 1 |
| HOPB | Fall | *Notemigonus crysoleucas* | 6 |
| KING | Spring | *Cyprinella lutrensis* | 2 |
| KING | Spring | *Etheostoma pseudovulatum* | 4 |
| KING | Spring | *Etheostoma tennesseense* | 1 |
| KING | Fall | *Lepomis macrochirus* | 1 |
| KING | Fall | *Luxilus cornutus* | 1 |
| KING | Fall | *Moxostoma pisolabrum* | 1 |
| KING | Fall | *Phoxinus erythrogaster* | 78 |
| LECO | Spring | *Campostoma anomalum* | 1 |
| LEWI | Fall | *Cyprinella spiloptera* | 1 |
| LEWI | Spring | *Etheostoma flabellare* | 1 |
| LEWI | Spring | *Lepomis cyanellus* | 1 |
| LEWI | Spring | *Lepomis macrochirus* | 4 |
| MAYF | Fall | *Elassoma zonatum* | 1 |
| MAYF | Fall | *Erimyzon oblongus* | 1 |
| MAYF | Spring | *Lepomis auritus* | 2 |
| MAYF | Spring | *Lepomis cyanellus* | 1 |
| MAYF | Fall | *Etheostoma histrio* | 1 |
| MAYF | Spring | *Lepomis macrochirus* | 3 |
| MAYF | Fall | *Lythrurus bellus* | 2 |
| MAYF | Spring | *Minytrema melanops* | 2 |
| MAYF | Fall | *Micropterus henshalli* | 4 |
| MAYF | Fall | *Micropterus warriorensis* | 1 |
| MAYF | Spring | *Moxostoma poecilurum* | 10 |
| MAYF | Fall | *Notropis stilbius* | 65 |
| MAYF | Spring | *Pteronotropis hypselopterus* | 1 |
| MCDI | Fall | *Catostomus commersonii* | 1 |
| MCDI | Fall | *Etheostoma nigrum* | 133 |
| MCDI | Fall | *Lepomis megalotis* | 3 |
| MCDI | Fall | *Pimephales vigilax* | 5 |
| PRIN | Fall | *Cyprinus carpio* | 1 |
| PRIN | Spring | *Micropterus salmoides* | 2 |
| PRIN | Spring | *Notropis volucellus* | 71 |
| PRIN | Spring | *Pimephales vigilax* | 1 |
| WALK | Fall | *Notropis atherinoides* | 1 |

**Table A3.** Species caught during either 2017–2019 bouts or the 2020–2022 bouts but not over three-year periods.

| Site | Species | Years Caught | Count |
|---|---|---|---|
| ARIK | *Ameiurus melas* | 2017–2019 | 16 |
| ARIK | *Etheostoma exile* | 2017–2019 | 206 |
| ARIK | *Fundulus zebrinus* | 2017–2019 | 20 |
| ARIK | *Lepomis cyanellus* | 2017–2019 | 203 |
| BLUE | *Ameiurus melas* | 2017–2019 | 1 |
| BLUE | *Lythrurus umbratilis* | 2017–2019 | 6 |
| BLUE | *Cyprinella camura* | 2020–2022 | 1 |
| BLUE | *Micropterus salmoides* | 2017–2019 | 1 |
| BLUE | *Micropterus punctulatus* | 2017–2019 | 1 |
| BLUE | *Nocomis asper* | 2017–2019 | 10 |
| BLUE | *Notropis buchanani* | 2020–2022 | 6 |
| BLUE | *Notropis nubilus* | 2020–2022 | 1 |
| BLUE | *Notropis suttkusi* | 2017–2019 | 61 |
| BLUE | *Notropis volucellus* | 2017–2019 | 99 |

**Table A3.** *Cont.*

| Site | Species | Years Caught | Count |
|------|---------|--------------|-------|
| BLUE | *Pimephales notatus* | 2017–2019 | 79 |
| BLUE | *Pylodictis olivaris* | 2017–2019 | 1 |
| CUPE | *Anguilla rostrata* | 2017–2019 | 3 |
| CUPE | *Gobiomorus dormitor* | 2020–2022 | 4 |
| CUPE | *Sicydium punctatum* | 2017–2019 | 30 |
| CUPE | *Sicydium plumieri* | 2017–2019 | 45 |
| GUIL | *Gambusia affinis* | 2017–2019 | 43 |
| GUIL | *Tilapia rendalli* | 2020–2022 | 1 |
| HOPB | *Ameiurus nebulosus* | 2017–2019 | 1 |
| HOPB | *Noturus gyrinus* | 2017–2019 | 1 |
| HOPB | *Salmo trutta* | 2017–2019 | 57 |
| KING | *Cyprinella lutrensis* | 2020–2022 | 2 |
| KING | *Etheostoma pseudovulatum* | 2017–2019 | 4 |
| KING | *Etheostoma tennesseense* | 2017–2019 | 1 |
| KING | *Luxilus cornutus* | 2020–2022 | 1 |
| KING | *Lepomis macrochirus* | 2017–2019 | 1 |
| KING | *Moxostoma pisolabrum* | 2020–2022 | 1 |
| KING | *Notropis percobromus* | 2020–2022 | 1 |
| KING | *Phoxinus erythrogaster* | 2017–2019 | 78 |
| KING | *Noturus exilis* | 2020–2022 | 2 |
| LEC0 | *Campostoma anomalum* | 2017–2019 | 1 |
| LEWI | *Gambusia holbrooki* | 2020–2022 | 46 |
| LEWI | *Lepomis cyanellus* | 2020–2022 | 1 |
| LEWI | *Nocomis leptocephalus* | 2020–2022 | 3 |
| LEWI | *Etheostoma flabellare* | 2017–2019 | 1 |
| LEWI | *Lepomis macrochirus* | 2017–2019 | 4 |
| MAYF | *Elassoma zonatum* | 2017–2019 | 1 |
| MAYF | *Lythrurus bellus* | 2017–2019 | 2 |
| MAYF | *Minytrema melanops* | 2017–2019 | 2 |
| MAYF | *Notropis ammophilus* | 2017–2019 | 4 |
| MAYF | *Erimyzon oblongus* | 2020–2022 | 1 |
| MAYF | *Etheostoma chlorosomum* | 2020–2022 | 2 |
| MAYF | *Etheostoma histrio* | 2020–2022 | 1 |
| MAYF | *Etheostoma nigrum* | 2020–2022 | 7 |
| MAYF | *Lepomis cyanellus* | 2020–2022 | 1 |
| MAYF | *Lepomis macrochirus* | 2020–2022 | 3 |
| MAYF | *Micropterus warriorensis* | 2020–2022 | 1 |
| MAYF | *Notropis volucellus* | 2020–2022 | 30 |
| MAYF | *Pteronotropis hypselopterus* | 2020–2022 | 1 |
| MCDI | *Ameiurus natalis* | 2020–2022 | 4 |
| MCDI | *Catostomus commersonii* | 2020–2022 | 1 |
| MCDI | *Notropis atherinoides* | 2017–2019 | 15 |
| MCDI | *Notropis shumardi* | 2017–2019 | 4 |
| MCDI | *Micropterus punctulatus* | 2017–2019 | 4 |
| MCDI | *Etheostoma nigrum* | 2017–2019 | 133 |
| POSE | *Cottus bairdii* | 2017–2019 | 1010 |
| PRIN | *Cyprinus carpio* | 2017–2019 | 1 |
| PRIN | *Micropterus salmoides* | 2017–2019 | 2 |
| PRIN | *Notropis stramineus* | 2017–2019 | 1 |
| PRIN | *Notropis volucellus* | 2017–2019 | 71 |
| PRIN | *Pimephales vigilax* | 2017–2019 | 1 |
| WALK | *Cottus caeruleomentum* | 2017–2019 | 55 |
| WALK | *Notropis atherinoides* | 2017–2019 | 1 |

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
