# Peer review of "Spatial Patterns in Fish Assemblages across the National Ecological Observation Network (NEON): The First Six Years"

_fishes, doi:10.3390/fishes8110552_

Round 1
Reviewer 1 Report
Comments and Suggestions for Authors
I completed my evaluation of the manuscript fishes-2539884 “Spatial and Temporal Patterns in Fish Assemblages Across the NEON Observatory After the First Six Years of Sampling” submitted to Fishes. This study addresses an interesting topic evaluation of relatively long-term data collected for continental database. The text structure is clear and well-written. In several places, I was lost, and I had to go back to find appropriate information. Especially, abbreviations are not used consistently. Please, see below specific points:
L8: I recommend using the whole network name in the abstract.
L11: I miss information, how the fish were collected? What method was applied?
L13: I recommend deleting the sentence “We calculated…”. It is clear from the following text.
L76: Please, explain the reason “NEON does not sample fish at the large river sites”.
L78: Please, explain the used protocol of sampling. I do not understand “sampling can occur anywhere in the littoral zone or the pelagic zone of the lake”. Do you mean random design? Are you not choosing representative places?
L86: Please, explain abbreviations, when mentioned first in the text. NOAA, MODIS and later in the text.
L93: Please, add details about electrofishing. Refer to a protocol, what device and settings were applied? Was it identical through the 6 years?
L121: I am not sure about the fish's wet weight. A subsample was weighted, and the rest counted. Was the whole catch weighted to get the total biomass from a locality?
L126: I get lost, to why other fishing methods “gill netting, and fyke netting counts” are mentioned here after detailed description of electrofishing. If the data were not used, exclude the methods from the text, if the other data were used, describe the fish sampling in more detail.
L128: I am very surprised by the catch per unit effort units: individuals per hour. I will expect individuals per m2 or hectare (area). How about biomass per area? Is it possible to use the common units?
L158/161: Please, here and elsewhere in the text. Introduce first the abbreviation (wet weight, ww) and late use it in the text “0.3 mg ww”.
L162: I miss information, how the fish size structure was evaluated?
L175-179: Here and elsewhere in the text, e.g. Table 4,5, L277, L292. Use italics when writing scientific (=Latin) names.
L186: I prefer to condensate the rows with 0 values. At least 0.000000 = simple 0.
L191: Please, add scale bars to the maps, it is difficult to compare the different US states.
L212: Please, explain the abbreviations Mass and Tenn in the Tabel caption.
L246: Is it possible to add numbers on the Y-axis and these numbers add to Tabel 5 to connect the species names between the Table and the Figure? With the exception of 6 species, the other species in Figure 7 are missing.
L271: I recommend merging the paragraphs.
L276: I think the family name should be “Salmonidae”, not Salmonid.
L342: The title of the article is missing.
L357: The reference style is entirely different from the others.
Author Response
L8: I recommend using the whole network name in the abstract.
(DM) changed
L11: I miss information, how the fish were collected? What method was applied? (DM) it is in section 2.3 NEON fish data
L13: I recommend deleting the sentence “We calculated…”. It is clear from the following text.
(DM) changed
L76: Please, explain the reason “NEON does not sample fish at the large river sites”. (DM) changed
L78: Please, explain the used protocol of sampling. I do not understand “sampling can occur anywhere in the littoral zone or the pelagic zone of the lake”. Do you mean random design? Are you not choosing representative places?
(DM) the lake part has been deleted as it is not relevant - so that should solve this problem
L86: Please, explain abbreviations, when mentioned first in the text. NOAA, MODIS and later in the text.
(DM) changed
L93: Please, add details about electrofishing. Refer to a protocol, what device and settings were applied? Was it identical through the 6 years?
(DM)I have put the protocol name and mentioned that it wasn’t changed significantly for the last 6 years –
L121: I am not sure about the fish's wet weight. A subsample was weighted, and the rest counted. Was the whole catch weighted to get the total biomass from a locality?
(DM)I put the word individual which should explain this question
L126: I get lost, to why other fishing methods “gill netting, and fyke netting counts” are mentioned here after detailed description of electrofishing. If the data were not used, exclude the methods from the text, if the other data were used, describe the fish sampling in more detail.
(DM) thisis just the name of the data product that is downloaded
L128: I am very surprised by the catch per unit effort units: individuals per hour. I will expect individuals per m2 or hectare (area). How about biomass per area (we don’t weigh every fish ? Is it possible to use the common units?
(DM)I believe time is an excepted CPUE and does not change the Alpha scores significantly
L158/161: Please, here and elsewhere in the text. Introduce first the abbreviation (wet weight, ww) and late use it in the text “0.3 mg ww”.
(DM)Done
L162: I miss information, how the fish size structure was evaluated?
(Added)
L175-179: Here and elsewhere in the text, e.g. Table 4,5, L277, L292. Use italics when writing scientific (=Latin) names.
(DM)done
L186: I prefer to condensate the rows with 0 values. At least 0.000000 = simple 0. (DM) changed
L191: Please, add scale bars to the maps, it is difficult to compare the different US states.
(DM) We want people to pay attention to the spatial spread not so much the states- please let me know if in that context it still makes sense to add a scale bar
L212: Please, explain the abbreviations Mass and Tenn in the Tabel caption. taken out (DM) dropped
L246: Is it possible to add numbers on the Y-axis and these numbers add to Tabel 5 to connect the species names between the Table and the Figure? With the exception of 6 species, the other species in Figure 7 are missing.
(DM) dropped
L271: I recommend merging the paragraphs.done
(DM) changed
L276: I think the family name should be “Salmonidae”, not Salmonid.
(DM) changed
L342: The title of the article is missing.
(DM) fixed
L357: The reference style is entirely different from the others.
(DM) changed
Reviewer 2 Report
Comments and Suggestions for Authors
Title: Spatial and Temporal Patterns in Fish Assemblages Across the NEON Observatory After the First Six Years of Sampling
General comments: The paper describes the analysis of 6 years of fish assemblage data collected from the NEON study in the United States. In general, I’m not really sure what your objective is in this paper. I’m not super sure if the work is supposed to be a first preliminary assessment of the data and its performance over its first 6 years and what the possible reasons this analysis was needed. Are there questions about the utility of this approach and what it can and cannot do? As it stands, the paper now seems to be a bit ‘data-dumpy’ – there’s lots of data shown, but that’s about it.
Additionally, the abstract is choppy and some material is in the wrong place; not all of the methods are described in the methods. The intro also jumps around a little. My suggestion is to first talk generally about fish assemblages and how these assessments might be done (like using alpha and beta diversity), but then also maybe about the problems NEON is designed to remedy (without mentioning it by name just yet). Then you can maybe get into NEON and why it was developed, but also some of the limitations, like scaling up from a few individual sites to a much larger area.
Specific comments:
Line 3: I’m always unsure about what to do with acronyms in titles. I’m sure you know it’s moderately controversial, but I think not using an acronym is the less offensive version. Just to be clear, I’m not saying don’t use an acronym, I’m saying you decide.
Line 8: is “network” really the expanded NEON acronym?
Line 8: saying the program is ‘thirty-year’ is sort of confusing. Has it been running for thirty years or is its lifespan 30 years? Open source suggests it is modifiable by the user. I think you mean ‘open access.’ I’m also not convinced calling it ‘continental-scale’ is appropriate.
Lines 11-12: so it has 24 more years to go?
Line 19: Beta test? Do you mean the usefulness of testing Beta diversity?
Lines 20-21: Why is more work required? And what does ‘scaling up’ mean? Do you mean scaling up from the individual sites to the continent? Or to the ecoregions shown in Figure 1?
Line 24: ‘NEON sites follow a predictable spatial pattern of fish assemblage’ is ambiguous. Did you make any predictions? Or are you referring to consistent assemblages of fishes at individual sites over time?
Lines 30-33: I think a missed point about a program like this is that researchers can easily get this sort of data on their own, but it’d likely have some constraints, like a small area, unique methods, and for a short time period. The benefit of a ‘background’ program like NEON is to overcome those sorts of challenges (but also at the expense of some specificity).
Lines 34-35: I think you mean ‘consistent observations’ can allow users to do these things and forms part of a larger approach which can often be neglected by individuals focussing on small areas for short periods.
Line 37: ‘;’ seems to be a typo
Line 42: some are highly mobile yes, but others less so
Line 44: mobility is an advantage if you are looking at regional influences (which you are), but it can be a major disadvantage if your focus is site-specific. It’s worth mentioning the difference so readers don’t freak out at this blanket statement.
Lines 45-46: this is a bit of a sore thumb – I have no reason to doubt this statement, but what does it have to do with your work?
Line 51: This statement is awkward. I think you mean a foundational principle of NEON is that fish assemblages can be indicators of anthropogenic influence.
Lines 52-54: this can be tightened up
Lines 55-58: This statement doesn’t seem to fit in this spot. It also made me reconsider this intro as a whole and inspired general comments about the structure of the introduction – for example, I think you got into NEON too early. First talking generally about fish assemblage data and how you can use it in environmental assessments might be a better place to start.
Lines 66-68: I think this is too jargony for the intro, especially since this terminology has not been introduced previously. For example, talking about the organization of the NEON sites, and a brief description of the logic used to select them might make this statement more understandable.
Figure 1: ok, so what are ‘core aquatic’ and ‘gradient aquatic’ sites? I didn’t see this described in lines 71-79. And what are the codes showing? Are these ecoregions? And if so, which level in the hierarchy? You also need a scale bar. Which of these sites are considered in your study? You may need to make a custom map for this paper to show only the sites with fish data.
Line 83: every year?
Line 86: missing open bracket
Lines 92-93: I don’t understand what this means. ‘Fish typically last’?
Line 97: typo
Line 109-110: This is out of place
Line 119: ok – and are these fish kept separate from the others during capture somehow? I can definitely see some bias in getting individuals out of a bucket for length and weight being a potential problem over time.
Line 120: what are the resolutions of the measurements? And what about fish with and without forked tails? You still use total length for fish with forks?
Lines 133-134: there seems to be missing information or some typos here
Line 136: only when your data are spatially sparse. I think you will have some serious issues with the scaling up of these data.
Line 137: Missing period.
Lines 142-147: why is this in the Beta diversity section? Besides, once you have enough data, you can include things like effort as a factor in the model (or you can use it to adjust your response metrics beforehand).
Lines 148-151: this should be in the alpha diversity section
Line 152: Beta
Line 154: need the citation for the vegan package
Line 155: permANOVA. You also need to define NMDS. Are these analyses being used to examine Beta diversity or something else? Please be clear about why each analysis was done and what information you wanted to get from them.
Lines 159-161: this needs to be stated above
Line 162: typo. Also, on line 110 you say 2017-2022
Line 165: but aren’t some fish not identified to species? I assume family or genus might be the lowest consistent level the identifications can get to? So the number of potential species is a range is it not?
Line 169: Lowest scores? Do you mean the lowest richness?
Lines 172-173: I don’t understand what these ‘start of growing season bout mean, the end of the growing season bout mean, highest score, lowest score’ are.
Table 2: I think you can cut some decimals from these scores. Again, what are ‘start’ and ‘end’ bouts? Is that the spring and fall sampling events? And these rows are sorted in descending order from highest to lowest Shannon Diversity? This needs to be stated in the table caption. This table is also way too big. These data are probably better represented in the main text in a figure (probably faceted by site and x=year) with this table being moved to the supplemental information
Figure 4: ok – what are these data? Are they the mean Shannon diversity indices for the sites? The maxima? The caption needs more detail
Lines 194-200: this belongs in the methods. It’s also filled with jargon – did you do some DNA sequencing on some of these fish? ‘0.2’ not ‘.2’ Why did you group by state? I thought ecoregion was your unit of analysis?
Lines 207-208: this needs to be in the methods
Lines 210-211: also needs to be in the methods
Figure 5: what are the ellipses?
Section 3.5: Is this described in the methods?
Table 4: genus and species need to be in italics or underlined. I’m also still confused about this start-end terminology
Lines 241-242: only if there isn’t bias in how the fish are chosen for measurements
Figure 7: Are the colours, symbol size, and x-axis all showing the same data?
Lines 261-263: ok so is this the purpose of your work? If so, this needs to be stated much earlier in the manuscript.
Lines 265-270: this probably needs to be in the intro
Line 272: ‘plays out’ is colloquial
Line 277: italics for genus and species
Lines 280-283: these are results
Line 284: saying ‘occurrence’ is cleaner
Line 292: this might end up being a big problem in your 30 year data set
Lines 306-309: in fairness, a more detailed study on this specific location using far more effort is probably a better option for this question. I think the point of NEON is to look at non-specific patterns in these fish communities that otherwise might be unknown because ‘academic’ science tends to disincentivize their study.
Lines 313-314: sure, but you didn’t assess future change. And anyone in ten years can use the existing data, and not this paper, to establish (or re-establish) the baseline.
Lines 318-320: did you define any expected fish assemblages? Or are you saying that over the first 6 years of NEON, the assemblage metrics were relatively stable?
Comments on the Quality of English Languagejust a few typos to fix
Author Response
Line 3: I’m always unsure about what to do with acronyms in titles. I’m sure you know it’s moderately controversial, but I think not using an acronym is the less offensive version. Just to be clear, I’m not saying don’t use an acronym, I’m saying you decide.
(DM) changed
Line 8: is “network” really the expanded NEON acronym?
(DM) changed
Line 8: saying the program is ‘thirty-year’ is sort of confusing. Has it been running for thirty years or is its lifespan 30 years? Open source suggests it is modifiable by the user. I think you mean ‘open access.’ I’m also not convinced calling it ‘continental-scale’ is appropriate.
(DM) clarified
Lines 11-12: so it has 24 more years to go?
(DM) clarified
Line 19: Beta test? Do you mean the usefulness of testing Beta diversity?
(DM) I have dropped the beta test I never felt like it was a great analysis and now I can test predicted continental scale distribution using alpha
Lines 20-21: Why is more work required? And what does ‘scaling up’ mean? Do you mean scaling up from the individual sites to the continent? Or to the ecoregions shown in Figure 1?
(DM) Dropped
Line 24: ‘NEON sites follow a predictable spatial pattern of fish assemblage’ is ambiguous. Did you make any predictions? Or are you referring to consistent assemblages of fishes at individual sites over time? (DM) clarified
Lines 30-33: I think a missed point about a program like this is that researchers can easily get this sort of data on their own, but it’d likely have some constraints, like a small area, unique methods, and for a short time period. The benefit of a ‘background’ program like NEON is to overcome those sorts of challenges (but also at the expense of some specificity).
(DM) added
Lines 34-35: I think you mean ‘consistent observations’ can allow users to do these things and forms part of a larger approach which can often be neglected by individuals focussing on small areas for short periods.
(DM) changed
Line 37: ‘;’ seems to be a typo
(DM) changed
Line 42: some are highly mobile yes, but others less so
(DM) changed
Line 44: mobility is an advantage if you are looking at regional influences (which you are), but it can be a major disadvantage if your focus is site-specific. It’s worth mentioning the difference so readers don’t freak out at this blanket statement.
(DM) removed as it wasn’t essential to my point
Lines 45-46: this is a bit of a sore thumb – I have no reason to doubt this statement, but what does it have to do with your work?
(DM) fair point I removed it
Line 51: This statement is awkward. I think you mean a foundational principle of NEON is that fish assemblages can be indicators of anthropogenic influence.
(DM) changed
Lines 52-54: this can be tightened up-
(DM) I believe I fixed this
Lines 55-58: This statement doesn’t seem to fit in this spot. It also made me reconsider this intro as a whole and inspired general comments about the structure of the introduction – for example, I think you got into NEON too early. First talking generally about fish assemblage data and how you can use it in environmental assessments might be a better place to start. (DM)
I believe I fixed this
Lines 66-68: I think this is too jargony for the intro, especially since this terminology has not been introduced previously. For example, talking about the organization of the NEON sites, and a brief description of the logic used to select them might make this statement more understandable.
(DM) removed
Figure 1: ok, so what are ‘core aquatic’ and ‘gradient aquatic’ sites? I didn’t see this described in lines 71-79. And what are the codes showing? Are these ecoregions? And if so, which level in the hierarchy? You also need a scale bar. Which of these sites are considered in your study? You may need to make a custom map for this paper to show only the sites with fish data.
(DM) fixed
Line 83: every year?
(DM) fixed
Line 86: missing open bracket
(DM) fixed
Lines 92-93: I don’t understand what this means. ‘Fish typically last’?
(DM) removed as it was unimportant
Line 97: typo
(DM) coud not find it
Line 109-110: This is out of place
(DM) moved
Line 119: ok – and are these fish kept separate from the others during capture somehow? I can definitely see some bias in getting individuals out of a bucket for length and weight being a potential problem over time.
(DM) We have checked for biases – and this does not seem to be a problem
Line 120: what are the resolutions of the measurements And what about fish with and without forked tails? You still use total length for fish with forks
(DM) yes
Lines 133-134: there seems to be missing information or some typos here
(DM) fixed
Line 136: only when your data are spatially sparse. I think you will have some serious issues with the scaling up of these data.
(DM) dropped
Line 137: Missing period.
(DM) dropped
Lines 142-147: why is this in the Beta diversity section? Besides, once you have enough data, you can include things like effort as a factor in the model (or you can use it to adjust your response metrics beforehand).
(DM) dropped
Lines 148-151: this should be in the alpha diversity section
(DM) not needed
Line 152: Beta
(DM) dropped
Line 154: need the citation for the vegan package
(DM) dropped
Line 155: permANOVA. You also need to define NMDS. Are these analyses being used to examine Beta diversity or something else? Please be clear about why each analysis was done and what information you wanted to get from them.
(DM) dropped
Lines 159-161: this needs to be stated above
(DM) dropped
Line 162: typo. Also, on line 110 you say 2017-2022
(DM) dropped
Line 165: but aren’t some fish not identified to species? I assume family or genus might be the lowest consistent level the identifications can get to? So the number of potential species is a range is it not?
Line 169: Lowest scores? Do you mean the lowest richness?
(DM) fixed
Lines 172-173: I don’t understand what these ‘start of growing season bout mean, the end of the growing season bout mean, highest score, lowest score’ are.
(DM) changed
Table 2: I think you can cut some decimals from these scores. Again, what are ‘start’ and ‘end’ bouts? Is that the spring and fall sampling events? And these rows are sorted in descending order from highest to lowest Shannon Diversity? This needs to be stated in the table caption. This table is also way too big. These data are probably better represented in the main text in a figure (probably faceted by site and x=year) with this table being moved to the supplemental information
(DM) decimal cut- I played around a bit with the other options and still believe that the table works best
Figure 4: ok – what are these data? Are they the mean Shannon diversity indices for the sites? The maxima? The caption needs more detail fixed
Lines 194-200: this belongs in the methods. It’s also filled with jargon – did you do some DNA sequencing on some of these fish? ‘0.2’ not ‘.2’ Why did you group by state? I thought ecoregion was your unit of analysis?
- DM) dropped
Lines 207-208: this needs to be in the methods
(DM) dropped
Lines 210-211: also needs to be in the methods
(DM) dropped
Figure 5: what are the ellipses?
(DM) dropped
Section 3.5: Is this described in the methods? dropped
Table 4: genus and species need to be in italics or underlined. I’m also still confused about this start-end terminology
(DM) changed
Lines 241-242: only if there isn’t bias in how the fish are chosen for measurements
(DM) we tested and found no biases
Figure 7: Are the colours, symbol size, and x-axis all showing the same data?
(DM) yes and I feel that is explained in the caption
Lines 261-263: ok so is this the purpose of your work? If so, this needs to be stated much earlier in the manuscript.
(DM) changed
Lines 265-270: this probably needs to be in the intro
(DM) rewrote
Line 272: ‘plays out’ is colloquial
(DM) changed
Line 277: italics for genus and species
(DM) changed
Lines 280-283: these are results
(DM) dropped
Line 284: saying ‘occurrence’ is cleaner
(DM) changed
Line 292: this might end up being a big problem in your 30 year data set
(DM) I agree
Lines 306-309: in fairness, a more detailed study on this specific location using far more effort is probably a better option for this question. I think the point of NEON is to look at non-specific patterns in these fish communities that otherwise might be unknown because ‘academic’ science tends to disincentivize their study.
(DM) modified
Lines 313-314: sure, but you didn’t assess future change. And anyone in ten years can use the existing data, and not this paper, to establish (or re-establish) the baseline.
(DM) Rewritten and changed with an emphasis on better hypothesis
Lines 318-320: did you define any expected fish assemblages? Or are you saying that over the first 6 years of NEON, the assemblage metrics were relatively stable?
(DM) Changed with dropping beta and using Alpha to test distribution at neon sites
Reviewer 3 Report
Comments and Suggestions for Authors
Comments:
This paper assessed spatial and temporal patterns in fish assemblages across the NEON observatory. The results suggest that fish size composition was stable at sites across time, and NEON sites follow predictable spatial diversity patterns. This paper deals with relevant and interesting topic and is potentially a valuable contribution to studies about fish ecology at large spatial scale. However, there are numerous problems that need to be addressed. The following is my concerns.
1. The authors need to better describe the specific objectives of this study. Otherwise, we do not know what this paper study for.
2. The authors need to provide the testable hypothesis and prediction we can follow for this study.
3. The authors need to better describe about the statistical analyses used for this study (e.g., more detailed explanation about NMDS and permaNOVA).
4. The discussion of this study seems to be out of focus possibly because the specific objectives are not well written in the introduction.
5. The management implication of this study is vague. I recommend that the authors write the more specific management implications based on your results in the conclusion.
Author Response
- The authors need to better describe the specific objectives of this study. Otherwise, we do not know what this paper study for. (DM) I have changed the paper so it tests the continental scale of the fish assemblages at NEON sites
- The authors need to provide the testable hypothesis and prediction we can follow for this study. (DM) see above
- The authors need to better describe about the statistical analyses used for this study (e.g., more detailed explanation about NMDS and permaNOVA). (DM) BETA test dropped I have some reservations about their value for this data
- The discussion of this study seems to be out of focus possibly because the specific objectives are not well written in the introduction. (DM) I hope that has changed with the rewrite
- The management implication of this study is vague. I recommend that the authors write the more specific management implications based on your results in the conclusion. (DM) I hope that has changed with the rewrite
Round 2
Reviewer 2 Report
Comments and Suggestions for Authors
Thank you for addressing the comments. The article has been improved.
Comments on the Quality of English LanguageThe paper does, however, require some minor editing (which can be addressed during production and I don't think should prevent it's acceptance). For example, the word 'observatory' in the title now seems redundant.
Author Response
Thank you for checking our manuscript.
Reviewer 3 Report
Comments and Suggestions for Authors
I think that the authors appropriately respond to my comments. I recommend that the authors check the manuscript again to make sure if there are minor mistakes in the manuscript. Thank you for giving me the opportunity for reviewing this manuscript.
Author Response
Thank you for checking our manuscript.